# Physiological Mechanisms of Titanium Regulation of Growth, Photosynthesis, and Mineral Absorption in Tartary Buckwheat

Anyin Qi [1,†], Zhengshan Wang [1,†], Liangzhen Jiang [1,†], Qiang Wang [1,2], Yuanhang Ren [1], Chenggang Liang [3], Yan Wang [3], Changying Liu [1], Xueling Ye [1], Yu Fan [1], Qi Wu [1], Xiaoyong Wu [1], Lianxin Peng [1], Dabing Xiang [1], Laichun Guo [2], Gang Zhao [1], Liang Zou [1], Jingwei Huang [4,*] and Yan Wan [1,*]

1   Key Laboratory of Coarse Cereal Processing, Ministry of Agriculture and Rural Affairs, College of Food and Biological Engineering, Chengdu University, Chengdu 610106, China; qianyin@stu.cdu.edu.cn (A.Q.); wangzhengshan@stu.cdu.edu.cn (Z.W.); jiangliangzhen@cdu.edu.cn (L.J.); wangqiangeternity@icloud.com (Q.W.); renyuanhang@cdu.edu.cn (Y.R.); liuchangying@cdu.edu.cn (C.L.); shirlingye@gmail.com (X.Y.); fandavi@163.com (Y.F.); jerviswuqi@126.com (Q.W.); wuxiaoyong@cdu.edu.cn (X.W.); penglianxin@cdu.edu.cn (L.P.); dabingxiang@163.com (D.X.); zhaogang@cdu.edu.cn (G.Z.); zouliang@cdu.edu.cn (L.Z.)
2   Baicheng Academy of Agricultural Sciences, No. 17, Sanhe Road, Taobei District, Baicheng 137099, China; guolaichun@126.com
3   Research Center of Buckwheat Industry Technology, School of Life Sciences, Guizhou Normal University, Guiyang 550001, China; jesselcg@163.com (C.L.); yanwanguf@163.com (Y.W.)
4   School of Preclinical Medicine, Chengdu University, Chengdu 610106, China
*   Correspondence: huangjingwei@cdu.edu.cn (J.H.); yanwan@cdu.edu.cn (Y.W.)
†   These authors contributed equally to this work.

**Abstract:** Titanium has been reported to have positive effects on crop growth and production in various species. However, the impact of titanium on the Tartary buckwheat crops has not yet been studied. Therefore, an experiment was conducted to investigate the effect of spraying different concentrations of ionic titanium on the growth, photosynthesis, and uptake of mineral nutrients in Tartary buckwheat. The results showed that the application of titanium significantly improved dry matter accumulation, internode diameter, main stem node, root length, root average diameter, root surface area, root volume, grains per plant, and weight of grains per plant. Additionally, chlorophyll and photosynthetic parameters showed improvement regardless of the concentration of titanium used. The study found that titanium accumulation was mainly in leaves. The content of titanium in leaves showed a significant positive correlation with K, Ca, Mg, Mn, Cu, Zn, and B. This suggests a potential synergistic relationship between titanium and minerals in Tartary buckwheat leaves. Furthermore, the study also observed a significant increase in the total accumulation of P, K, Ca, Mg, Mn, Cu, Zn, and B in Tartary buckwheat plants. Overall, this study provides evidence for the positive effects of titanium on Tartary buckwheat and offers a theoretical foundation for practical production.

**Keywords:** titanium; plant growth; photosynthesis; Tartary buckwheat; mineral accumulation

## 1. Introduction

Tartary buckwheat (*Fagopyrum tataricum* (L.) Gaertn.) is a significant grain crop with medicinal and edible properties, playing essential roles in daily diets, traditional medicine, and economic development worldwide [1,2]. It is rich in nutrients and diverse bioactive phytochemicals, making it highly suitable for human consumption and promoting good health [3–5]. However, despite its immense market potential, the production of Tartary buckwheat remains inadequate to meet demand [6]. This is mainly due to the challenging growing environment including semi-arid, high altitude, infertile soil areas, etc., which are not suitable for cultivating main crops [7]. Additionally, the changing climate conditions further affect the yield and quality of Tartary buckwheat [8]. Therefore, it is crucial

to explore and implement new agricultural strategies and technologies to enhance the production of Tartary buckwheat and other crops.

Titanium (Ti) is a transition metal element that ranks ninth in abundance in the Earth's crust. It is widely used in various industries, including pigments, cosmetics, medical devices, electronics, and agriculture, due to its unique properties [9,10]. In the field of sustainable agriculture, titanium shows great potential as a photocatalytic material, biostimulant, and pesticide for remediating soil contamination, promoting plant germination and growth, and controlling plant pathogenic microbes [11]. The positive impact of titanium on crop production has been demonstrated in various studies. It has been found to be absorbed and utilized by crops such as wheat (*Triticum aestivum* L.) [12], maize (*Zea mays* L.) [13], tomatoes (*Solanum lycopersicum* L.) [14], and lettuce (*Lactuca sativa* L.) [15], resulting in improved plant growth, crop productivity, and fruit quality [16]. These positive effects may come from improved photosynthesis, mineral element absorption etc., so titanium is also known as a beneficial element for plants [17]. In rice (*Oryza sativa* L.), titanium treatment enhances the net photosynthetic rate by increasing photochemical quenching, decreasing non-photochemical fluorescence quenching, and regulating the sugar metabolism pathway to promote growth [18]. Titanium treatment increases chlorophyll content and expression of the light-harvesting complex II (LHC–II) gene, thereby improving light absorption and energy transfer and promoting oxygen release [19,20]. Larger leaf area and thickness as well as thicker chloroplast grana stacks and larger starch granules were observed in titanium-treated soybean leaves [21]. Additionally, studies have found that the application of titanium improves nitrogen (N) uptake and photo-assimilation in tomato plants, especially under low-nitrogen conditions [22]. In low-phosphorus (P) environments, titanium application enhances the P content and increases the levels of Ca, Mg, Cu, Zn, and Fe in wheat plants [23]. Under hydroponic conditions, the application of titanium promotes the contents of microelements N, P, K, Ca, and Mg in tomato leaves [24]. In soybean (*Glycine max* (L.) Merr.) plants, the application of ionic titanium enhances phosphorus uptake by soybeans by changing the rhizosphere system (including root morphology, root exudates, and rhizosphere pH), thereby increasing plant utilization of phosphorus [25]. In addition, the application of titanium enhances the ability of plants to tolerate extreme environments, including salt, drought, high-temperature, and heavy metal pollution conditions [26–29].

In summary, the application of titanium has the effect of promoting plant photosynthesis and nutrient absorption, improving plant production, and enhancing plant stress resistance [30]. So far, the application of titanium in buckwheat has not been reported. Therefore, we hypothesized that spraying ionic titanium could improve the growth, photosynthetic parameters, mineral element absorption, and yield of Tartary buckwheat plants. This will improve the production capacity of Tartary buckwheat in harsh environments. The findings of this research will serve as a theoretical basis for utilizing titanium in Tartary buckwheat production.

## 2. Materials and Methods

### 2.1. Materials and Plant Growth

In this study, we used the Tartary buckwheat cultivar ChaunQiao No. 1, which is extensively grown in southwest China. The stable ionic titanium solution, provided by Tigrow (Tianjin) Science and Technology Ltd. Tianjin, China, has a titanium concentration of $3.5 \text{ g·L}^{-1}$ and does not contain any metallic elements or amino acids.

This experiment was conducted at the Training Cultivation Experiment Station of Chengdu University in Chengdu City, Sichuan Province, China (latitude 30°39′ N, longitude 104°11′ E, altitude 491 m), during the Tartary buckwheat growing season from September to November 2022 and March to June 2023. Ten healthy seeds were planted in pots with dimensions of 25 cm in diameter and 20 cm in height. The pots contained a soil mixture of natural and organic manure in a 3:1 ratio, and each pot was filled with 5 kg of this soil mixture. After 20 days of sowing, the seedlings were thinned to three plants per pot. The titanium treatment began when the seedlings reached the five-leaf stage. In 2022, different

concentrations of titanium (control (CK) = 0, 3.5, 7, 14, 28, 56, and 112 mg·L$^{-1}$) were sprayed on the leaves after 5 p.m. on a sunny day. Each treatment consisted of 20 pots. The second spray was applied on the eighth day after the first spray. Based on the results from 2022, titanium concentrations of CK, 28, and 56 mg·L$^{-1}$ were selected for verification. Samples were measured and collected at the anthesis and maturity stages.

### 2.2. Dry Matter Accumulation and Root System Parameters

The Tartary buckwheat plant was separated into leaves, stems, and roots; subjected to curing at 105 °C for 15 min; and subsequently dried at 65 °C until reaching a constant weight. A 20 × 20 cm square was used to dig down 30 cm with the plant at the center. Then, we carefully removed the soil around the roots, followed by gently running water to remove the soil attached to the root surface and subsequently rinsing three times with distilled water. The root system was captured using a root scanner (Epson Expression 12000XL, Japan), and the parameters of the root system (total root length, root surface area, root volume, and average root diameter) were analyzed using WinRHIZO Pro 32-bit 2017a software (Regent Instruments Inc., Quebec, QC, Canada).

### 2.3. Chlorophyll Content

A fresh sample of 0.2 g was extracted in 80% acetone and subsequently filtered. The filtrate was utilized to measure the chlorophyll content detected with a spectrophotometer at 645 nm, 643 nm, and 470 nm and then used to calculate chlorophyll a (Chl a), chlorophyll b (Chl b), carotenoids, and total chlorophyll (Total Chl) by the method of Lichtenthaler [31].

### 2.4. Photosynthetic Parameters

Photosynthetic parameters were measured using a portable Gas–Exchange System GFS–3000 (WALZ Inc., Effeltrich, Germany) on a sunny day between 9:00–12:00 a.m. Net photosynthesis (Pn), intercellular $CO_2$ concentration (Ci), stomatal conductance (gs), and transpiration (Tr) were obtained.

### 2.5. Essential Elements of Plant Nutrients and Ti Analyses

Essential elements of plant as well as titanium were determined using the method described by Hussain et al. [21]. Samples were ground to a powder and passed through a 100-mesh sieve. The powder (0.1 g) was digested into colorless solution with 10 mL of nitric acid using an intelligent digestion instrument DS32–260 (CIF, Beijing, China). The colorless solution was diluted with 5% nitric acid and then filtered through 0.2 μm membrane filter paper. The final filter solution was analyzed using inductively coupled plasma–mass spectroscopy (ICP–MS; iCAP RQ, Thermo Fisher Scientific, Waltham, MA, USA) for quantification of titanium (Ti), potassium (K), magnesium (Mg), calcium (Ca), manganese (Mn), copper (Cu), boron (B), as well as zinc (Zn). The phosphorus concentration was measured spectrophotometrically, following the method described by Murphy et al. [32].

### 2.6. Agronomic Traits

Agronomic traits were measured following the method outlined by Xiang et al. [1]. From each treatment, twelve plants that exhibited uniform growth were harvested at maturity (with at least 70% of grains mature) to record the number of main stem nodes, 1- to 2-internode diameter, branch number of the main stem, and grain number per plant. A grain sample per plant was oven-dried at 35 °C for 72 h, and subsequently, the grain weight per plant, along with the 1000-grain weight, was determined.

### 2.7. Statistical Analyses

Data were collected and processed using Microsoft Excel 2016 (Microsoft Crop, Montgomery, AL, USA) and IBM SPSS Statistics 25 software (IBM, Chicago, AL, USA). After passing the normality and homogeneity tests, we utilized the Tukey multiple-range test in

one-way ANOVA analysis of variance for significance analysis ($p < 0.05$). The plots were drawn using GraphPad Prism 10.0 (GraphPad Software Corp, San Diego, CA, USA).

## 3. Results

### 3.1. Effect of Titanium on Agronomic Traits

As shown in Table 1, the analysis of the 2022 data revealed that the treatment with titanium had no significant effect on the height of the Tartary buckwheat plant, the internode diameter, the number of main stem branches, and the weight of 1000 grains. However, number of main stem nodes, grains per plant, and grain weight per plant initially increased and then decreased with the increase in ionic titanium concentration. The highest values were observed at a concentration of 56 mg·$L^{-1}$ $Ti^{4+}$, resulting in a significant increase of 7.69%, 31.69%, and 50.82%, respectively, compared to the CK treatment. Similarly, the analysis of the 2023 data indicated a significant increase in the internode diameter, number of main stem nodes, grains per plant, and grain weight per plant at the 56 mg·$L^{-1}$ $Ti^{4+}$ treatment. The increments were 9.57%, 5.8%, 46.25%, and 40.5%, respectively, compared to the CK treatment.

**Table 1.** Effect of titanium on agronomic traits of Tartary buckwheat.

| Year | Treatment | Plant Height (cm) | Internode Diameter (mm) | Main Stem Node (Number) | Main Stem Branch (Number) | Grain per Plant (Number) | Grain Weight per Plant (g) | 1000-Grain Weight (g) |
|---|---|---|---|---|---|---|---|---|
| 2022 | CK | 55.7 ± 1.54 ab | 4.73 ± 0.09 ab | 13 ± 0.12 bb | 6.75 ± 0.35 a | 151.7 ± 7.05 b | 3.05 ± 0.11 b | 20.5 ± 0.35 a |
| | 3.5 mg·$L^{-1}$ $Ti^{4+}$ | 58.88 ± 1.49 ab | 4.85 ± 0.14 ab | 12.83 ± 0.17 b | 6.5 ± 0.29 a | 178.5 ± 11.46 b | 3.6 ± 0.22 ab | 19.74 ± 0.55 a |
| | 7 mg·$L^{-1}$ $Ti^{4+}$ | 57.69 ± 2.07 ab | 4.45 ± 0.09 b | 13.08 ± 0.15 b | 6.67 ± 0.28 a | 160.7 ± 15.5 b | 2.98 ± 0.26 b | 19.8 ± 0.61 a |
| | 14 mg·$L^{-1}$ $Ti^{4+}$ | 60.25 ± 1.23 a | 4.57 ± 0.1 ab | 13.33 ± 0.19 ab | 6.08 ± 0.29 a | 177 ± 10.73 b | 3.55 ± 0.29 b | 19.49 ± 0.74 a |
| | 28 mg·$L^{-1}$ $Ti^{4+}$ | 58.07 ± 1.39 ab | 4.8 ± 0.12 ab | 13.08 ± 0.26 b | 6.5 ± 0.34 a | 195.6 ± 14.67 ab | 4.03 ± 0.29 ab | 21.47 ± 0.51 a |
| | 56 mg·$L^{-1}$ $Ti^{4+}$ | 55.17 ± 0.97 ab | 5.1 ± 0.09 a | 14 ± 0.21 a | 6.75 ± 0.18 a | 222.1 ± 14.2 a | 4.6 ± 0.22 a | 20.29 ± 0.6 a |
| | 112 mg·$L^{-1}$ $Ti^{4+}$ | 54.08 ± 1.11 b | 4.7 ± 0.13 ab | 13.5 ± 0.19 ab | 6.83 ± 0.27 a | 163.6 ± 9.51 b | 3.52 ± 0.21 b | 21.33 ± 0.6 a |
| 2023 | CK | 67.58 ± 0.63 a | 6.27 ± 0.06 b | 15.33 ± 0.25 b | 7.61 ± 0.11 a | 307.22 ± 15.44 b | 6.37 ± 0.28 b | 20.73 ± 0.19 a |
| | 28 mg·$L^{-1}$ $Ti^{4+}$ | 71.86 ± 1.72 a | 6.95 ± 0.06 a | 15.94 ± 0.15 ab | 8.22 ± 0.22 a | 395.56 ± 46.57 ab | 8.01 ± 0.91 ab | 20.54 ± 0.21 a |
| | 56 mg·$L^{-1}$ $Ti^{4+}$ | 71.49 ± 5.02 a | 6.87 ± 0.12 a | 16.22 ± 0.15 a | 8.22 ± 0.15 a | 449.32 ± 6.42 a | 8.95 ± 0.13 a | 20.18 ± 0.07 a |

The data in the table represent the mean ± SE, and different letters indicate significant differences between different treatments in the same year, according to the Tukey significance test ($p < 0.05$). The control is an equal amount of distilled water without ionic titanium, defined as CK.

### 3.2. Effect of Titanium on Dry Matter Accumulation

The application of titanium enhanced the dry matter accumulation of Tartary buckwheat. In a two-year pot experiment conducted in 2022 and 2023, the dry matter accumulation of buckwheat leaves, stems, and roots showed an increase across various treatment concentrations (Table 2). In 2022, the increases ranged from 16.47% to 34.65%, 23.66% to 42.38%, and 16.46% to 29.88%. In 2023, the corresponding increases were 15.54% to 18.24%, 7.35% to 10.05%, and −2.08% to 12.56%. The highest values for leaves and stems in both years were observed in the 56 mg·$L^{-1}$ $Ti^{4+}$ treatment, while the maximum value for roots was found in the 28 mg·$L^{-1}$ $Ti^{4+}$ treatment. Conversely, the lowest values for leaves, stems, and roots were identified in the CK treatment.

**Table 2.** Effect of titanium on dry matter accumulation of Tartary buckwheat.

| Year | Treatment | Leaf (g) | Stem (g) | Root (g) |
|---|---|---|---|---|
| 2022 | CK | 1.8 ± 0.11 b | 1.38 ± 0.1 b | 0.174 ± 0.007 b |
| | 3.5 mg·$L^{-1}$ $Ti^{4+}$ | 2.14 ± 0.06 ab | 1.7 ± 0.1 ab | 0.212 ± 0.008 ab |
| | 7 mg·$L^{-1}$ $Ti^{4+}$ | 2.25 ± 0.11 a | 1.77 ± 0.14 ab | 0.214 ± 0.016 a |
| | 14 mg·$L^{-1}$ $Ti^{4+}$ | 2.22 ± 0.14 a | 1.77 ± 0.12 ab | 0.202 ± 0.02 ab |
| | 28 mg·$L^{-1}$ $Ti^{4+}$ | 2.09 ± 0.08 ab | 1.92 ± 0.14 ab | 0.225 ± 0.006 a |
| | 56 mg·$L^{-1}$ $Ti^{4+}$ | 2.37 ± 0.1 a | 1.96 ± 0.17 a | 0.215 ± 0.008 a |
| | 112 mg·$L^{-1}$ $Ti^{4+}$ | 2.09 ± 0.06 ab | 1.86 ± 0.13 ab | 0.220 ± 0.009 a |

**Table 2.** *Cont.*

| Year | Treatment | Leaf (g) | Stem (g) | Root (g) |
|---|---|---|---|---|
| | CK | 3.09 ± 0.01 b | 2.17 ± 0.03 b | 0.3 ± 0.008 b |
| 2023 | 28 mg·L$^{-1}$ Ti$^{4+}$ | 3.57 ± 0.15 a | 2.33 ± 0.03 ab | 0.34 ± 0.002 a |
| | 56 mg·L$^{-1}$ Ti$^{4+}$ | 3.65 ± 0.02 a | 2.39 ± 0.06 a | 0.3 ± 0.007 b |

The data in the table represent the mean ± SE, and different letters indicate significant differences between different treatments in the same year, according to the Tukey significance test ($p < 0.05$). The control is an equal amount of distilled water without ionic titanium, defined as CK.

### 3.3. Effect of Titanium on Root Growth

Varying the concentration of titanium treatment had a positive impact on the growth of buckwheat roots (Table 3). In the 2022 experiment, there was an initial increase in total root length, root surface area, and root volume, followed by a decrease as the titanium concentration increased. When compared to the control group, the 7–28 mg·L$^{-1}$ Ti$^{4+}$ treatment showed significant improvements in total root length, the 14 mg·L$^{-1}$ Ti$^{4+}$ treatment showed significant improvements in root surface area, the 56 mg·L$^{-1}$ Ti$^{4+}$ treatment showed significant improvements in average root diameter (AvgDiam), and the 14–28 mg·L$^{-1}$ Ti$^{4+}$ treatment showed significant improvements in root volume. In the 2023 experiment, compared to the control, both the 28 and 56 mg·L$^{-1}$ Ti$^{4+}$ treatments significantly increased the total root length and root surface area, the 56 mg·L$^{-1}$ Ti$^{4+}$ treatment significantly increased the average root diameter, and the 28 mg·L$^{-1}$ Ti$^{4+}$ treatment significantly increased the root volume. The maximum root average diameter value in both years was observed under the 56 mg·L$^{-1}$ Ti$^{4+}$ treatment.

**Table 3.** Effect of titanium on root growth of Tartary buckwheat.

| Year | Treatment | Length/cm | Surf Area/cm$^2$ | AvgDiam/mm | Root Volume/cm$^3$ |
|---|---|---|---|---|---|
| | CK | 257.74 ± 8.83 b | 33.32 ± 1.75 b | 0.41 ± 0.008 b | 0.34 ± 0.023 b |
| | 3.5 mg·L$^{-1}$ Ti$^{4+}$ | 312.67 ± 13.49 ab | 39.42 ± 1.75 ab | 0.42 ± 0.01 ab | 0.41 ± 0.02 ab |
| | 7 mg·L$^{-1}$ Ti$^{4+}$ | 331.74 ± 20.65 a | 42.53 ± 2.93 ab | 0.42 ± 0.011 ab | 0.44 ± 0.035 ab |
| 2022 | 14 mg·L$^{-1}$ Ti$^{4+}$ | 322.92 ± 12.88 a | 43.68 ± 1.57 a | 0.44 ± 0.01 ab | 0.48 ± 0.022 a |
| | 28 mg·L$^{-1}$ Ti$^{4+}$ | 327.2 ± 23.62 a | 42.85 ± 3.71 ab | 0.43 ± 0.012 ab | 0.47 ± 0.039 a |
| | 56 mg·L$^{-1}$ Ti$^{4+}$ | 269.04 ± 9.78 ab | 39.16 ± 1.71 ab | 0.45 ± 0.008 a | 0.45 ± 0.02 ab |
| | 112 mg·L$^{-1}$ Ti$^{4+}$ | 263.59 ± 9 ab | 34.81 ± 1 ab | 0.44 ± 0.011 ab | 0.39 ± 0.015 ab |
| | CK | 430.44 ± 4.06 b | 70.12 ± 0.34 c | 0.47 ± 0.005 b | 0.85 ± 0.026 b |
| 2023 | 28 mg·L$^{-1}$ Ti$^{4+}$ | 540.92 ± 21.26 a | 83.4 ± 2 a | 0.49 ± 0.002 ab | 1.03 ± 0.048 a |
| | 56 mg·L$^{-1}$ Ti$^{4+}$ | 490.92 ± 10.06 a | 74.53 ± 0.84 b | 0.51 ± 0.01 a | 0.96 ± 0.009 ab |

The data in the table represent the mean ± SE, and different letters indicate significant differences between different treatments in the same year, according to the Tukey significance test ($p < 0.05$). The control is an equal amount of distilled water without ionic titanium, defined as CK.

### 3.4. Effect of Titanium on Leaf Chlorophyll

According to the findings in Figure 1, the titanium treatment increased photosynthetic pigments in Tartary buckwheat leaves. In 2022, the data demonstrated that chlorophyll a, chlorophyll b, and total chlorophyll showed an increase in the 3.5–28 and 112 mg·L$^{-1}$ Ti$^{4+}$ treatments. Based on the 2023 data, chlorophyll a, chlorophyll b, carotenoids, and total chlorophyll exhibited significant increases under the 28 and 56 mg/L Ti treatments compared to the CK treatment.

### 3.5. Effects of Titanium on Photosynthetic Parameters

The results from both the 2022 and 2023 experiments demonstrate that titanium spraying had a significant impact on Tartary buckwheat photosynthetic parameters (Figure 2). In 2022, the Pn showed a significant increase in treatments with titanium concentrations of 3.5 and 14–56 mg·L$^{-1}$ Ti$^{4+}$. Additionally, compared to the CK treatment, intercellular carbon dioxide concentration (Ci), transpiration rate (Tr), and stomatal conductance (Gs) exhibited an increase in the 7–112 mg·L$^{-1}$ Ti$^{4+}$ treatment but did not reach a significant

level, whereas they exhibited a significant increase in the 3.5 mg·L$^{-1}$ Ti$^{4+}$ treatment, respectively. In 2023, like 2022, the Pn showed a significant increase at 28 and 56 mg·L$^{-1}$ Ti$^{4+}$ treatments compared to the control group, while Gs and Tr significantly increased at the 28 mg·L$^{-1}$ Ti$^{4+}$ treatment.

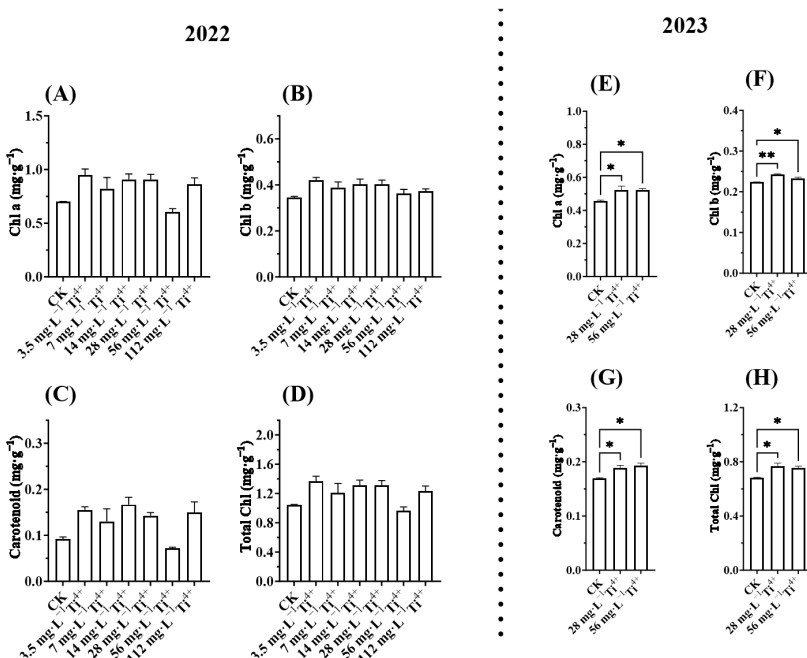

**Figure 1.** The impact of titanium on chlorophyll levels in Tartary buckwheat. (**A**,**E**) represent chlorophyll a; (**B**,**F**) denote chlorophyll b; (**C**,**G**) stand for carotenoids; and (**D**,**H**) represent total chlorophylls. The data in the columns represent the mean $\pm$ SE, and * and ** indicate statistical significance at $p < 0.05$ and $p < 0.01$, respectively. The control is an equal amount of distilled water without ionic titanium, defined as CK.

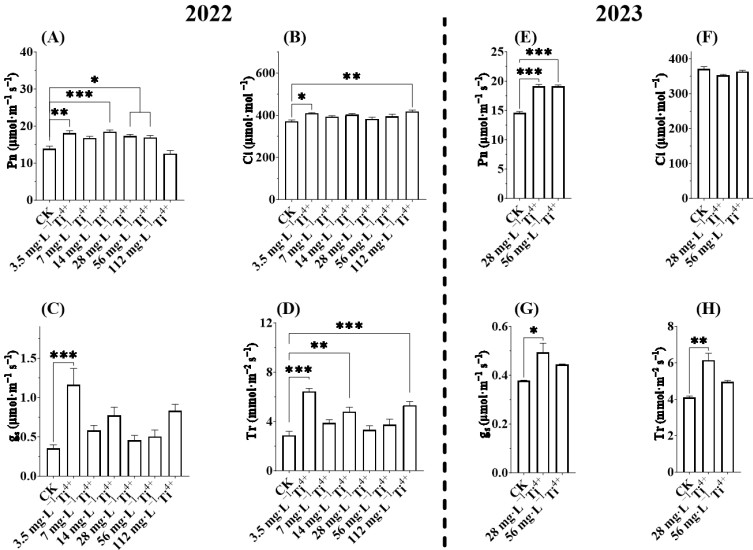

**Figure 2.** The impact of titanium on photosynthetic parameter in Tartary buckwheat. (**A**,**E**) represent the net photosynthetic rate; (**B**,**F**) denote intercellular carbon dioxide concentration; (**C**,**G**) stand for stomatal conductance; and (**D**,**H**) represent the transpiration rate. The data in the columns represent the mean $\pm$ SE, and *, **, and *** indicate statistical significance at $p < 0.05$, $p < 0.01$, and $p < 0.001$, respectively. The control is an equal amount of distilled water without ionic titanium, defined as CK.

### 3.6. Titanium Uptake and Distribute

As depicted in Figure 3, after applying titanium through foliar spraying, the main accumulation of titanium was observed in the leaves. There was no significant difference in the titanium content between stems and roots compared to the CK. Additionally, the titanium content in leaves and the overall titanium accumulation in plants showed an upward trend with increasing titanium concentration.

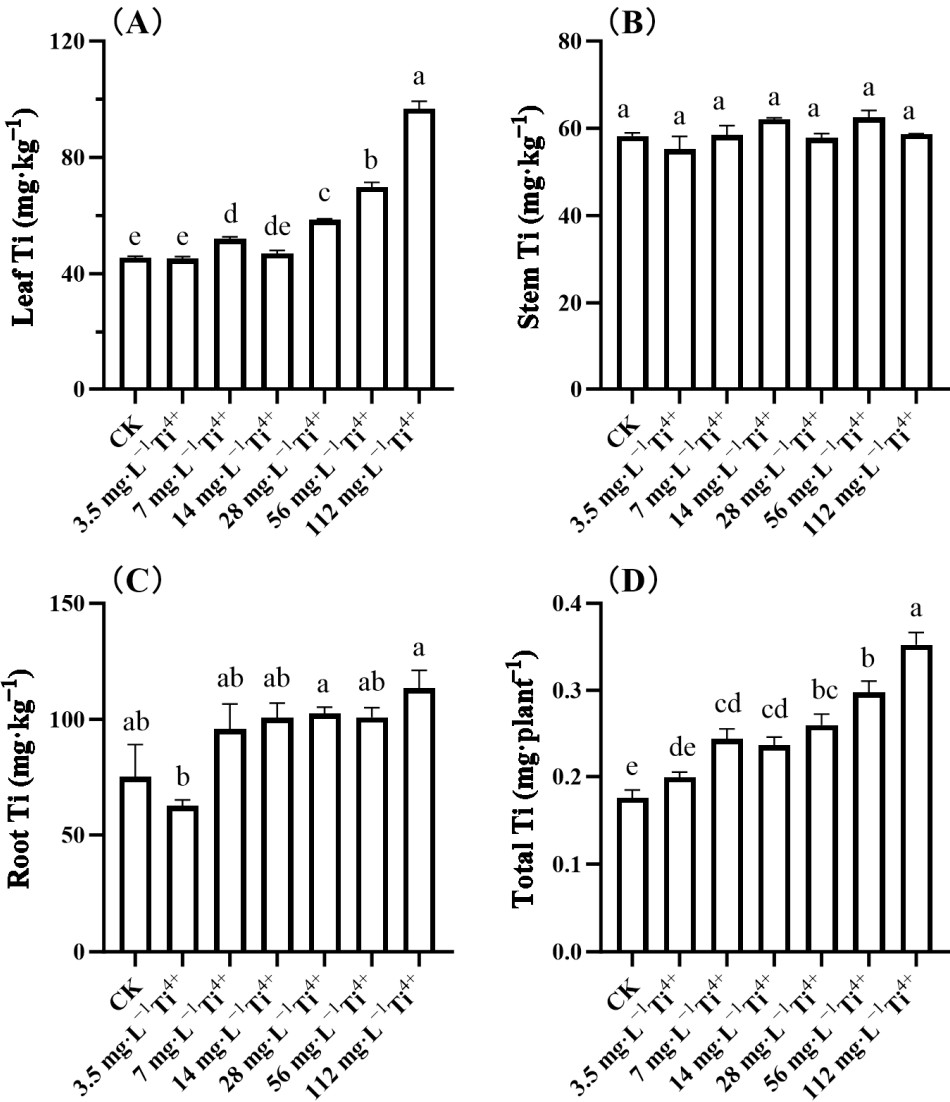

**Figure 3.** Distribution of titanium in various organs, namely (**A**) leaves, (**B**) stems, and (**C**) roots, and (**D**) total plant accumulation. The data in the table represent the mean ± SE, and different letters indicate significant differences between different treatments in the same year, according to the Tukey significance test (*p* < 0.05). The control is an equal amount of distilled water without ionic titanium, defined as CK.

### 3.7. Correlation between Titanium Content and Mineral Element Content in Leaves

Linear regression analysis indicated a slight positive correlation between titanium and phosphorus (P) content (Figure 4A). Additionally, the content of potassium (K), magnesium (Mg), calcium (Ca), manganese (Mn), copper (Cu), zinc (Zn), and boron (B) exhibited significant positive correlation with titanium in Tartary buckwheat leaf (Figure 4B–H).

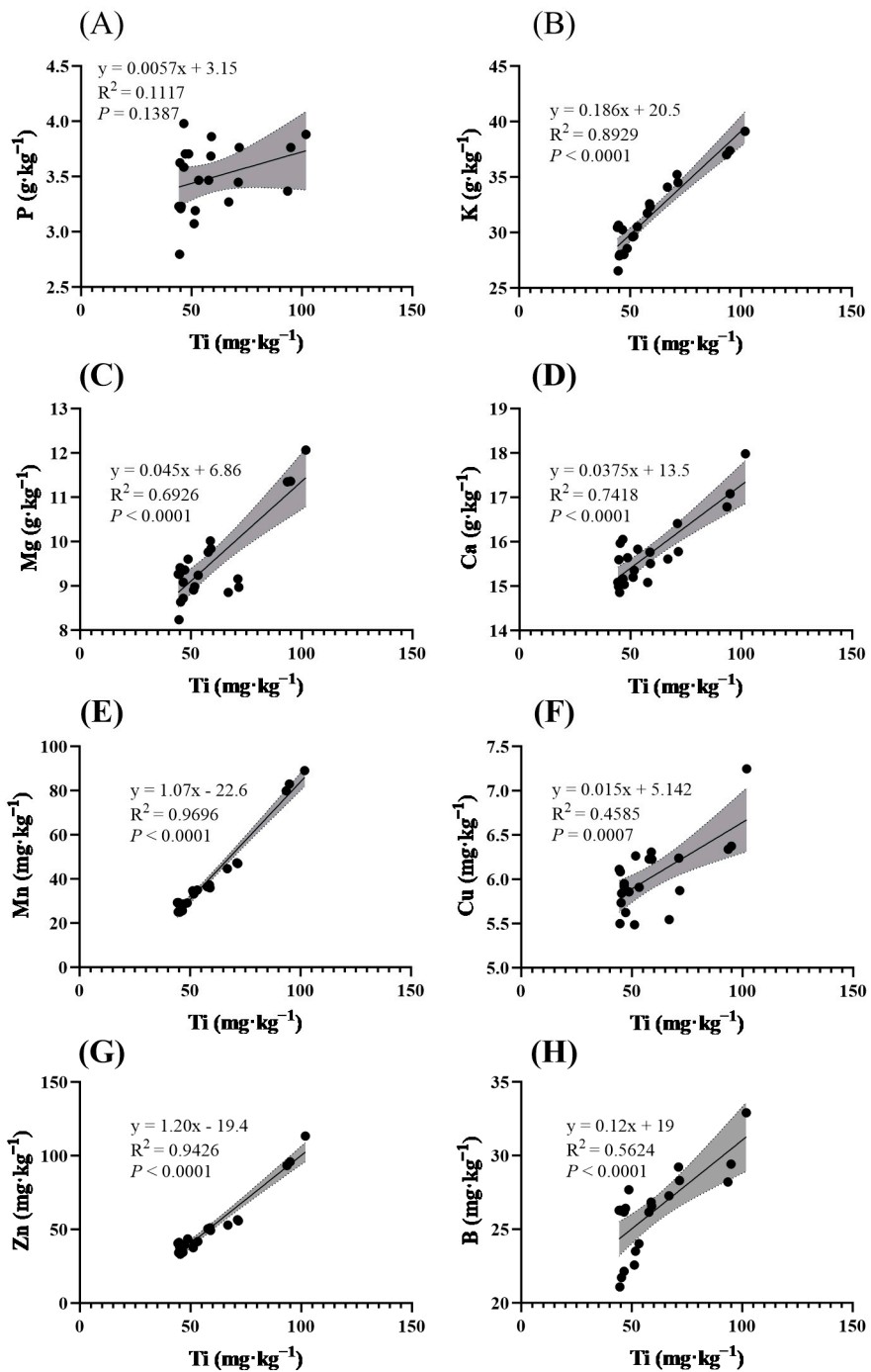

**Figure 4.** Correlations between leaf titanium (Ti) and macroelements (**A–D**) as well as microelements (**E–H**) were examined. Simple linear regression fits are represented by solid lines, with 95% confidence intervals shown in grey. A *p*-value of less than 0.05 was considered to indicate a significant correlation.

### 3.8. Effects of Titanium on Mineral Elements Accumulation in Plant

To assess the nutritional status of Tartary buckwheat under varying concentrations of titanium, we analyzed the accumulation of macro- and microelements (Figure 5), Figure 5A–D show the accumulation of macroelements in Tartary buckwheat at different concentrations of titanium. We observed a higher accumulation of P in the titanium treatment, with an increase ranging from 23.15% to 42.27%. The highest values were recorded in the 56 mg·L$^{-1}$ Ti$^{4+}$ treatment. The concentration of K, Mg, and Ca also showed a gradual increase in response to the titanium concentration, with increments of

29.12–60.21%, 28.88–63.4%, and 12.64–37.77%, respectively. Additionally, the accumulation of microelements (Mn, Cu, Zn, and B) significantly increased after titanium treatment, with increases of 22.96–188.68%, 19.81–42.8%, 41.14–165.37%, and 25.23–51.33%, respectively (Figure 5E–H).

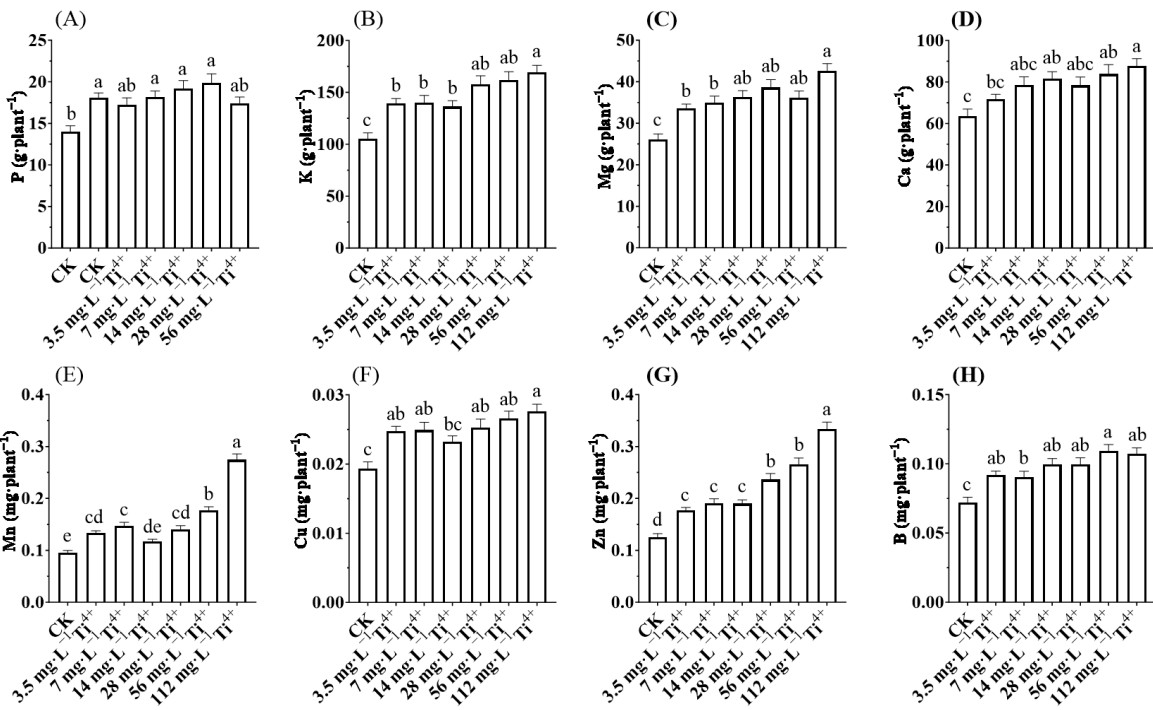

**Figure 5.** The impact of titanium on the accumulation of mineral elements in Tartary buckwheat plants (**A–D**). Macroelements P, K, Mg, and Ca are indicated by letters (**E–H**), respectively, while microelements Mn, Cu, Zn, and B are indicated by letters (**E–H**), respectively. The data in the table represent the mean $\pm$ SE, and different letters indicate significant differences between different treatments in the same year, according to the Tukey significance test ($p < 0.05$). The control is an equal amount of distilled water without ionic titanium, defined as CK.

## 4. Discussion

### 4.1. The Plant Growth Was Stimulated by Titanium

Titanium is a plant-benefitting element that has been shown to enhance plant growth and crop yields. Various types of titanium compounds, including organic titanium, nanotitanium (TiO$_2$ NPs), and ionic titanium (Ti$^{4+}$), have demonstrated these benefits [17]. For example, Tytanit$^{®}$, a commercial biostimulant containing organic titanium compounds, effectively improved plant biomass and crop yield in winter oilseed rape (*Brassica napus* L.) [33], *Festulolium* hybrid (*Festulolium braunii* (K. Richt) A. Camus) [34], soybean (Glycine max (L.) Merr.) [35], hybrid alfalfa (*Medicago sativa* L.), and red clover (*Trifolium pratense* L.) [36]. TiO$_2$NPs, a nanomaterial widely used in various industries, hold great potential in agricultural production [30]. Low concentrations of nanotitanium promote crop growth, enhance photosynthetic parameters, and facilitate nutrient absorption, but high concentrations may exhibit toxic effects [37,38]. As a form more readily absorbed by plants, ionized titanium promotes the enhancement of plant growth parameters and improves the shade tolerance of soybeans (*Glycine max* (L.) Merr.) [39] and Chinese hickory (*Carya cathayensis* Sarg.) [40] after foliar spraying. In this study, the application of titanium showed positive effects on plant growth and yield of Tartary buckwheat (Tables 1–3). The dry matter accumulation (Table 2) and root growth (Table 3) significantly increased when treated with titanium. The highest values were observed at 56 mg·L$^{-1}$ Ti$^{4+}$ for dry matter accumulation and 28 mg·L$^{-1}$ Ti$^{4+}$ for root growth. This is possibly attributed to titanium stimulating auxin accumulation in the roots, enhancing antioxidant enzymes, and boosting photosynthetic

capacity [41,42]. The enhanced growth of plant roots and leaves provides the plant with additional resources, ultimately improving agronomic traits of Tartary buckwheat, such as internode diameter, main stem nodes, grains per plant, and grain weight per plant (Table 1). The application of titanium resulted in an improved internode diameter and main stem node, which may enhance collapse resistance. This speculation originated from observations in soybean plants under intercropping treated with ionic titanium, where higher stem strength and lignin content were recorded [43], but systematic studies are needed to confirm this. The maximum values of the grain number per plant and the grain weight per plant increased by 46.41% and 37.06%, respectively, when exposed to 56 mg·L$^{-1}$ Ti$^{4+}$. Consistent with our research results, 50 mg·L$^{-1}$ TiO$_2$NPs treatment significantly increased the number of seeds per plant and yield in sweet corn (*Zea mays* L.) [44], and 125–250 mg·L$^{-1}$ Ti$^{4+}$ treatment significantly increased the seeds number per plant, the seed weight per plant, and yield in soybean (*Glycine max* (L.) Merr.) [43]. Furthermore, the use of titanium has been found to stimulate pollination and fertilization in apples (*Malus pumila* Mill.), litchi (*Litchi chinensis* Sonn.), and blueberry (*Vaccinium* spp.), resulting in increased fruit set rate and fruit yield [45–47]. However, further research is required to establish whether the application of titanium directly promotes crop pollination and fertilization.

### 4.2. Plant Photosynthesis Was Improved by Titanium

Chlorophyll plays a crucial role in the plant photosynthetic system, as it absorbs and transmits light energy to drive photosynthesis [48]. The application of titanium has been found to upregulate the expression patterns of genes associated with the light-harvesting complex ll and the tetrapyrrole ring in chlorophyll, thereby promoting chlorophyll synthesis [19,49]. In our study, we observed a significant increase in chlorophyll content upon the application of titanium (Figure 1). This enhanced absorption and transfer of solar energy directly impacts the photosynthetic activity [50]. Similarly, a positive effect on photosynthetic parameters was observed. The changes in Pn, Ci, Gs, Tr, and chlorophyll all exhibited the same trend, showing non-concentration-dependent increases (Figures 1 and 2). This phenomenon has also been reported in cultivated radish (*Raphanus sativus* L.), where the increase in Pn is not dependent on the dosage [51]. The beneficial effects of titanium on chlorophyll and photosynthetic parameters have been documented in other plant species such as vetiver (*Vetiveria zizanioides* L. Nash) [52], *Ulmus elongate* (*Ulmus elongata* L. K. Fu & C. S. Ding) [53], and Chinese hickory (*Carya cathayensis* Sarg.) [40]. In the presence of 28 and 56 mg·L$^{-1}$ Ti$^{4+}$, Pn was still significantly increased compared to the control, but Ci, Gs, and Tr showed no significant difference from the control. This may be attributed to the improvement of instantaneous water-use efficiency, electron transport rate, and rubisco enzyme activity in the photosynthetic system [39,51]. At the same time, titanium treatment resulted in a larger leaf area, providing greater sunlight exposure and photosynthetic site area [21]. Additionally, the presence of titanium affected the uptake in Tartary buckwheat leaf of minerals associated with photosynthesis (Figure 4), including P, Mn, and Mg. P is a crucial component of ADP and ATP, nucleic acids, and phospholipids in the chloroplast [54]. Mn plays an essential role in the oxygen-evolving complex in photosystem II [55]. Furthermore, the combination of magnesium ions and tetra porphyrin rings is a vital component of chloroplasts [56]. The combined effects of these factors contribute to the overall enhancement of photosynthesis [57].

### 4.3. Titanium Promotes Mineral Element Absorption

Understanding the accumulation of titanium in plants is crucial for comprehending its impact on plants. Previous studies have demonstrated that TiO$_2$ NPs with a size below 36 nm can be absorbed through the roots and subsequently transferred to the shoots [58]. In this study, titanium was found to predominantly accumulate in leaves, with no significant changes in stems and roots (Figure 3). However, Sajad H. et al. [58] observed a significant increase in titanium content in roots and stems with treatments at 125 mg·L$^{-1}$ Ti$^{4+}$ and above. Further research is needed to elucidate the mechanisms of titanium transfer

within the plant and to understand the titanium requirements of different plant species. Furthermore, we observed that the increase in titanium levels in leaves was accompanied by an increase in mineral elements. The analysis of linear regression revealed a significant positive correlation between the titanium content in leaves and the levels of macro- and micronutrients such as Ca, Mg, Mn, Cu, Zn, and B (Figure 4). It is suspected that there is a synergistic effect between titanium and other mineral elements in plants. Previous studies have shown that there is a synergy between macro- and microelements through the exchange of signals from multiple elements [59]. The increase in titanium content in the leaves resulted in elevated levels of macro- and microelements, which in turn supported the growth, metabolism, and photosynthesis of the leaves. This finding is consistent with a study on soil-aged $TiO_2$ NPs nanoparticles that showed enhanced levels of Fe in leaves and Mg, Ca, Zn, and K in the roots of carrots [60]. In Arabidopsis, titanium treatment led to higher levels of P, K, Na, Fe, Mn, Cu, Zn, and S [61]. The accumulation of mineral elements in the plant significantly increased with titanium treatment, including P, K, Ca, Mg, Mn, Cu, Zn, and B (Figure 5). This increase can be attributed to the enhanced root structure, namely root length, root diameter, root volume, and root surface area (Table 3). The improved root system structure allows for better nutrient uptake, especially under nutrient deficiency [62]. Under low-P and shade-stress conditions, foliar application of titanium improves root auxin content, leading to increased P uptake [25]. Additionally, titanium application can promote root exudation, thereby increasing P availability [63]. When low concentrations of $TiO_2$ NPs are mixed with soil, they enhance the availability of soil N, P, Cu, Fe, and Mn and stimulate the growth of microbial species [64]. This evidence highlights the potential of titanium to enhance the efficiency of fertilizer use, particularly nitrogen and phosphorus fertilizers, which are commonly used in large quantities in production and known to cause significant environmental issues.

## 5. Conclusions

This study demonstrates the positive effects of ionic titanium on Tartary buckwheat, specifically in terms of plant growth, photosynthesis, and mineral absorption. The highest values for dry matter accumulation and yield of Tartary buckwheat were observed with a treatment of 56 mg·L$^{-1}$ $Ti^{4+}$. The chlorophyll levels and photosynthetic rate also increased under titanium treatment regardless of concentration. Titanium primarily accumulates in the leaves, and the titanium content in the leaves showed a significant positive correlation with K, Ca, Mg, Mn, Cu, Zn, and B. Additionally, there was a significant increase in the accumulation of both macro- and microelements. In summary, this study provides theoretical support for practical field production. The selected concentration of ionic titanium in our study can be directly used in practical Tartary buckwheat production to increase Tartary buckwheat yield.

**Author Contributions:** A.Q., writing—original draft, conceptualization, data curation, investigation, and formal analysis; L.J. and Z.W., formal analysis, writing—review and editing, visualization, investigation, and data curation; Q.W. (Qiang Wang) and Y.R., resources, investigation, methodology, and visualization; C.L. (Chenggang Liang) and Y.W. (Yan Wang), funding acquisition, resources, data analysis, and formal analysis; X.Y. and Y.F., resources, validation, and visualization, and investigation; X.W., C.L. (Changying Liu) and Q.W. (Qi Wu), resources, validation, and visualization, and data curation; D.X., supervision, formal analysis, resources, and funding acquisition; L.P. and L.G., supervision, formal analysis, resources, and methodology; G.Z. and L.Z., supervision, formal analysis, resources, and conceptualization; J.H. and Y.W. (Yan Wan), conceptualization, funding acquisition, project administration, editing, and validation. All authors have read and agreed to the published version of the manuscript.

**Funding:** This work was supported by the Science & Technology Department of Sichuan Province (2022YFQ0041, 2022NSFSC1725, 2023NSFSC0214, and 2023YFN0062); Open Project of the Key Laboratory of Panxi Characteristic Crop Research and Utilization at Xichang University (SZKF2209); China Agriculture Research System (CARS–07–B–1); The National Natural Science Foundation of China (32160428); Innovative Training Program for College Students (202311079040); and Undergraduate education and teaching reform project of Chengdu University (cdjgb2022186).

**Data Availability Statement:** All datasets supporting the conclusions of this article are included within the article. If not included in the manuscript, data are available from the corresponding author upon reasonable request.

**Conflicts of Interest:** The authors declare no conflicts of interest.

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
