# Peer review of "Physiological Mechanisms of Titanium Regulation of Growth, Photosynthesis, and Mineral Absorption in Tartary Buckwheat"

_agronomy, doi:10.3390/agronomy14040720_

Round 1
Reviewer 1 Report
Comments and Suggestions for Authors
It seems that in Figure 3 (D) on top of bar corresponding to 112 Ti, goes a "a" instead of "c"
Reviewer 2 Report
Comments and Suggestions for Authors
There are some major comments and concerns indicated in the reviewed document. Specially, authors must improve the edition of some texts. The species must include both the common names and the scientific names. The statistical analysis of data must be substantially improved. Since some inconsistences were detected in citations within the main texts, it is assumed that the Reference section must be carefully reviewed.

The are some words wrongly used. Some sentences are also confusing. The units and symbols must be written separately. No sentence may start with a number or an abbreviation. All abbreviations need to be defined the first time that is used. In general, the document may benefit from a proofreading by an native speaker expert in academic writing and editing
Reviewer 3 Report
Comments and Suggestions for Authors
The study investigates the impact of titanium on Tartary buckwheat, focusing on its growth, photosynthesis, and mineral nutrient uptake. While titanium has been previously reported to enhance crop growth, its effects on Tartary buckwheat's medicinal and food properties were not explored. The experiment involved spraying different titanium concentrations, revealing that titanium significantly improved various growth parameters, chlorophyll levels, and photosynthetic efficiency. Titanium accumulation was primarily in Tartary buckwheat leaves, with a positive correlation with essential minerals. The study also noted a significant increase in the total accumulation of several minerals in Tartary buckwheat plants. The findings suggest a potential synergistic relationship between titanium and minerals, offering theoretical support for practical production. However, the study has some limitations, such as not addressing the potential environmental impact of titanium application and lacking comparison with other relevant studies on titanium's effects on crops.
The research on the impact of titanium on Tartary buckwheat is novel in its focus on a specific crop and its potential medicinal and food properties. While previous studies have highlighted the positive effects of titanium on crop growth, this research uniquely explores its implications for a particular plant species. Overall, the research contributes new insights to the existing body of knowledge, shedding light on the specific benefits of titanium for Tartary buckwheat.
Relevant sources are cited in the Introduction. I would recommend adding a brief objective of the work and working hypotheses at the end of the chapter.
The Materials and Methods chapter contains the necessary information about the experiment. A description of the locality, the varieties used, evaluation methods during the vegetation, and subsequent analyses are presented.
It is clear from the results that in the first year 6 combinations with different treatment titanium were used, while in the following year, only two concentrations were verified. This narrowing is explained, but from a methodological point of view, it is not the correct procedure. The experiment should be repeated under the same conditions in both years. The presented results thus appear disproportionate, when data from 6 and two treatments are presented in the tables and graphs. It is to be considered whether to modify the results presented in the basic tables and graphs in both concentrations in both years and to present the additional concentrations (3.5, 7, 17, 112) as a supplement.
The discussion part can be considered well-crafted. An explanation of what the practical use of titanium plant treatment will be should be added to the conclusions.
Round 2
Reviewer 3 Report
Comments and Suggestions for Authors
Manuscript was improved based on my comments. The authors submitted fairly detailed explanations on comments and recommendations. It can be stated that they have added or explained everything necessary. In its modified form, the manuscript is suitable for publication.
